# A Non-Invasive Neonatal Signature Predicts Later Development of Atopic Diseases

**DOI:** 10.3390/jcm11102749

**Published:** 2022-05-12

**Authors:** Youssouf Sereme, Moïse Michel, Soraya Mezouar, Cheick Oumar Guindo, Lanceï Kaba, Ghiles Grine, Thibault Mura, Jean-Louis Mège, Tu Anh Tran, Pierre Corbeau, Anne Filleron, Joana Vitte

**Affiliations:** 1IHU Méditerranée Infection, 13005 Marseille, France; seremeyoussouf@yahoo.fr (Y.S.); moise0michel@gmail.com (M.M.); soraya.mezouar@univ-amu.fr (S.M.); cheicko86@gmail.com (C.O.G.); lancekaba@yahoo.fr (L.K.); grineghiles@gmail.com (G.G.); jean-louis.mege@univ-amu.fr (J.-L.M.); 2IRD, APHM, MEPHI, Aix-Marseille Université, 13284 Marseille, France; 3Immunology Department, University Hospital Nîmes, 30900 Nîmes, France; 4IRD, AP-HM, SSA, VITROME, Aix-Marseille Université, 13284 Marseille, France; 5UFR Odontologie, Aix-Marseille Université, 13284 Marseille, France; 6INSERM, University of Montpellier, U1061, Neuropsychiatry: Epidemiological and Clinical Research, 34093 Montpellier, France; thibault.mura@chu-nimes.fr; 7Laboratoire de Biostatistique, Epidémiologie Clinique, Santé Publique Innovation et Méthodologie (BESPIM), Groupe Hospitalier Caremeau, CHU de Nîmes, Nîmes University Hospital, 30900 Nîmes, France; 8Paediatrics Department, University Hospital Nîmes, 30900 Nîmes, France; tu.anh.tran@chu-nimes.fr; 9INSERM U1183, Institute for Regenerative Medicine & Biotherapy, 34295 Montpellier, France; 10Faculty de Medicine, Montpellier University, 34000 Montpellier, France; 11CNRS UMR 9002, Institute of Human Genetics, 34090 Montpellier, France; 12IDESP, INSERM UMR UA11, Institut Desbrest d’Epidemiologie et de Santé Publique (IDESP) Campus Sante, 34093 Montpellier, France

**Keywords:** preterm birth, fecal mediator and cytokine, methanogenic Archaea, allergy, atopy

## Abstract

**Background:** Preterm birth is a major cause of morbidity and mortality in infants and children. Non-invasive methods for screening the neonatal immune status are lacking. Archaea, a prokaryotic life domain, comprise methanogenic species that are part of the neonatal human microbiota and contribute to early immune imprinting. However, they have not yet been characterized in preterm neonates. **Objective:** To characterize the gut immunological and methanogenic Archaeal (MA) signature in preterm neonates, using the presence or absence of atopic conditions at the age of one year as a clinical endpoint. **Methods:** Meconium and stool were collected from preterm neonates and used to develop a standardized stool preparation method for the assessment of mediators and cytokines and characterize the qPCR kinetics of gut MA. Analysis addressed the relationship between immunological biomarkers, Archaea abundance, and atopic disease at age one. **Results:** Immunoglobulin E, tryptase, calprotectin, EDN, cytokines, and MA were detectable in the meconium and later samples. Atopic conditions at age of one year were positively associated with neonatal EDN, IL-1β, IL-10, IL-6, and MA abundance. The latter was negatively associated with neonatal EDN, IL-1β, and IL-6. **Conclusions:** We report a non-invasive method for establishing a gut immunological and Archaeal signature in preterm neonates, predictive of atopic diseases at the age of one year.

## 1. Introduction

Preterm birth, defined as delivery at fewer than 37 completed weeks of gestation, is the leading cause of neonatal mortality and morbidity and has long-term adverse health consequences [1]. The global incidence of preterm births was estimated at 10.6 in 2014, and 9.8 in 2000 [2]. The etiology of preterm birth is multifactorial and not yet fully understood [3]. However, factors related to preterm birth include maternal or fetal medical conditions, genetic and epigenetic [4] influences, environmental exposures, infertility treatments, behavioral and socioeconomic elements [3,5]. Preterm infants experience abnormal immune and metabolic programming, which might exert a lasting influence on the risk of future disease [6,7]. Preterm-born children have been shown to have immune mediator dysregulation [8], impaired innate immunity and adaptive responses characterized by reduced levels of immunoglobulin (Ig) G, opsonization and phagocytosis, and increased activation of Th1 cells compared to that of Th2 cells [9]. Cohort studies show that preterm-born children are at increased risk for preschool wheezing and school-age asthma [10], but not for food allergy [11,12] or atopic dermatitis (AD) [11,13].

During fetal life, maternal microbiota produces compounds that are transferred to the fetus and enhance the generation of innate immune cells [14]. This process is halted prematurely in preterm infants, leaving them vulnerable to disease [9]. Preterm infants have an inflammatory and hypoxic state, which has a negative impact on lung maturation, the risk of respiratory infections, and susceptibility to subsequent exposures [9,14,15]. As early as the neonatal period, the gut microbiota imprints a persistent effect on the immune system through multiple mechanisms, including the modulation of epithelial functions, the production of cytokines, and the recruitment and training of immune cells [16,17,18].

Archaea, considered a separate domain of life from Eukarya, giant bacteria and viruses, are part of the human microbiota [19,20,21]. Gut methanogenic Archaea consume hydrogen produced by bacterial fermentation, releasing methane and short chain fatty acids (SCFA) and thus taking part in the energy supply to the host [19]. They interact with the host immune system, triggering innate and adaptive immune responses, generation of specific T and B cells, and hypersensitivity responses in animals and humans [22,23,24]. We have shown that the neonatal gut is colonized by methanogenic Archaea from the first postnatal hours, possibly starting in utero [21,22]. Gut microbiome establishment is altered in preterm and low- birth-weight infants [25,26].

Clinical investigations and research studies in neonates, including those born before term, are usually performed with peripheral blood. However, the search for non-invasive alternatives has gained momentum in recent years [27,28].

We hypothesize the existence of an association between intestinal methanogenic Archaea and the intestinal immunological signature, understood as a pattern of immune biomarkers including cell-specific products, e.g., mast cell tryptase and eosinophil-derived neurotoxin (EDN) and major pro- and anti-inflammatory cytokines. The validation of this hypothesis would open the prospect of a predictive score for the later occurrence of immune disorders, including atopic diseases.

We addressed this question through the development of a non-invasive standardized method for the assessment of the neonatal gut immune and microbial status, implemented in a cohort of preterm infants. The aims of the present study were: (1) establish a non-invasive method adequate for the investigation of preterm neonates, (2) characterize the gut immune and Archaeal components longitudinally from birth to six weeks in the study cohort, and (3) correlate the results of gut immune and Archaeal investigations at birth and up to six weeks to the later occurrence of allergic or atopic conditions. 

## 2. Methods

### 2.1. Patients and Sampling

Stool samples from 43 preterm neonates were collected without the use of preservatives at the Nimes and Montpellier University Hospitals and stored at −80 °C. Samples were collected as meconium (n = 33) and later stool samples at 2 (n = 33), 4 (n = 29), and 6 (n = 24) weeks.

### 2.2. Ethics Statement

#### 2.2.1. A-Immunological Analysis

##### Preparation of Fecal Samples

One gram of feces was solubilized in two milliliters of an in-house extraction buffer consisting of phosphate buffered saline supplemented with 4 mM 4-(2-aminoethyl)-benzensulphonyl fluoride, 0.26 mM bestatin, 28 µM E-64, 2 µM leupeptin and 0.6 µM aprotinin, pH 7.4, and a protease inhibitor (Sigma-Aldrich, St. Louis, MN, USA [29]. The stool–buffer mixture was incubated for 20 min at room temperature, prior to centrifugation at 2000 rpm for 15 min at 4 °C. The supernatant liquids were freeze-dried for 24 h, resolubilized in one milliliter of extraction buffer, and used for mediator and cytokine determination (Figure 1).

##### Total IgE, Tryptase, and EDN Determination

The concentration of total IgE, tryptase, and EDN were measured using an automated fluoro-enzymo-immunoassay with the ImmunoCAP™ 250 platform (Thermo Fisher Scientific, Uppsala, Sweden), according to ISO 15,189 standards [30]. The measurement range was 2–5000 kIU/L (4.8–12,000 µg/L) for total IgE, 1–200 µg/L for tryptase, and 2–200 µg/L for EDN.

##### Calprotectin and Total Protein Determination

Fecal calprotectin was measured using the BIOFLASH (Werfen, Barcelona, Spain) chemo-luminescent analyzer platform according to ISO 15,189 standards. Assay sensitivity was greater than 20 µg/mL. The total protein concentration of the samples was measured by the colorimetric method (BCA Protein assay, Thermo Fisher Scientific).

##### Immunoassays

Cytokines (IL-6, IL-10, IL-1β, TGF-β, and TNF-α) were measured by ELISA using specific immunoassay kits according to the manufacturer’s protocols (R&D systems, Minneapolis, MN, USA). The sensitivity of the assays was 1.0 pg/mL.

#### 2.2.2. A-Microbiological Analysis: Methanogenic Archaea by qPCR

##### DNA Extraction and PCR Assays

For DNA extraction, 0.2 g of each stool sample were mixed in 1.5 mL tubes with 500 µL of G2 lysis buffer from an EZ1^®^DNA Tissue Kit (QIAGEN, Hilden, Germany). Then, 0.3 g of acid-washed beads ≤ 106 μm (Sigma-Aldrich, Saint Quentin Fallavier, France) were added in each tube and shaken in a FastPrep BIO 101 device (MP Biomedicals, Illkirch, France) for 45 s for mechanical lysis before 10 min incubation at 100 °C. A 180 µL volume of the mixture was then incubated with 20 µL of proteinase K (QIAGEN, Hilden, Germany) at 56 °C overnight before a second mechanical lysis was performed. Total DNA was finally extracted with an EZ1 Advanced XL extraction kit (QIAGEN) and 50 μL eluted volume. Sterile phosphate buffered saline (PBS) (Fisher Scientific, Illkirch, France) was used as a negative control in each DNA extraction run. Extracted DNA was incorporated into real-time PCR performed using Metha_16S_2_MBF: 5′-CGAACCGGATTAGATACCCG-3′ and Metha_16S_2_MBR: 5′-CCCGCCAATTCCTTTAAGTT-3′ primers and the FAM_Metha_16S_2_MBP 6FAM- CCTGGGAAGTACGGTCGCAAG probe targeting the 16S DNA gene of methanogens, designed in our laboratory (Eurogentec, Angers, France) as previously described [31]. PCR amplification was done in 20 μL volume including 15 μL of mix and 5 μL of extracted DNA. Five microliters of ultra-pure water (Fisher Scientific, Illkirch, France) were used instead of DNA in the negative controls. The amplification reaction was performed in a CFX96 thermocycler (BioRad, Marnes-la-Coquette, France) incorporating a protocol with a cycle of 50 °C for 2 min, followed by 39 cycles of 95 °C for 45 s, 95 °C for 5 s and finally 60 °C for 30 s. Samples with a CT < 40 were considered positive. Gene amplification and PCR sequencing were performed as previously described [25,26,32,33,34].

##### Statistical Analysis

The responses for each quantitative parameter were described using median and 25–75 percentile (interquartile range, IQR) unless otherwise stated. Analyses were performed using the Wilcoxon test when two groups were compared, and the Kruskal–Wallis test when more than two groups were compared. The association between the different biomarkers of interest were analyzed, at each sampling time, using Spearman’s correlation coefficient. The association profiles between different biomarkers were also analyzed using a principal component analysis method. Statistical analyses were performed at the conventional two-tailed α level of 0.05, using R 2.13.2 statistical software (R Foundation for Statistical Computing, https://www.r-project.org (accessed on 18 March 2022), Vienna, Austria).

## 3. Results

### 3.1. Demographic and Clinical Characteristics of Preterm Infants

The 43 preterm neonates included in our study had at birth an average weight of 1160.41 g (range 440–1750 g), an average gestational age of 29 weeks (range 24–32 weeks) and an average height of 37.11 cm (range 32–47 cm). Thirty-five (81%) were born by cesarean section and 8 (19%) by vaginal delivery. Only five mothers (11.6%) had received antibiotic therapy during the peripartum period. As part of the cohort follow-up, clinical evaluation (AF) was conducted at 1 year and assessed the presence or absence of health conditions, including atopic diseases. When necessary, allergy diagnosis was carried out according to current recommendations [35,36]. A total of nine children developed an atopic condition during the first year, manifested as asthma or cow’s milk allergy (CMA) in eight and AD in three, with two patients presenting an association of AD, asthma, and CMA (Table 1).

### 3.2. Immune Profiling

#### 3.2.1. Total Protein Determination

First, we measured the total protein content in all samples. The median concentration of fecal proteins was stable from birth to six weeks, ranging from 4.53 to 9.18 g/L (*p* = 0.10; Kruskal–Wallis) (Table 2).

#### 3.2.2. Immune Cell Markers and Cytokines

Total IgE was detectable in over 90% of the samples at all ages, in increasing amounts between birth (meconium) and six weeks (*p* < 0.0001; Kruskal–Wallis).

Conversely, tryptase detection increased with sampling age, reaching 58% in samples at six weeks, up from less than 15% at earlier times (*p* < 0.0001; Chi-square). As most values were lower than the quantification limit, quantitative comparison was not significant (*p* = 0.61, Kruskal–Wallis) (Figure 2 and Table 2). 

Calprotectin and EDN were detected in all samples at comparable levels irrespective of age (*p* = 0.13 and 0.21, Kruskal–Wallis) (Figure 2 and Table 2).

All cytokines except TNF-α were detectable in meconium and fecal samples. TGF-β and IL-6 were the most prevalent, detected in up to 90% of samples, while IL-10 was the less prevalent, found in 20% or less of the fecal samples. The frequency of detection of IL-6 and IL-1β increased with age (*p* < 0.0001; chi-square), although there was a sharp drop in IL-6 frequency of detection and measured levels between meconium (75%, median 11.6 pg/L) and samples at two weeks (21%, median 0.2 pg/L).

TGF-β and IL-1β median concentrations increased with age (*p* = 0.014 and 0.001, respectively; Kruskal–Wallis). The median level of IL-6 was the highest in meconium samples and increased again at six weeks (*p* < 0.0001; Kruskal–Wallis). IL-10 median concentrations did not vary with age (Figure 2 and Table 2).

Maternal antibiotic therapy and route of delivery did not significantly affect the meconium levels of cytokines, total IgE, tryptase, calprotectin, and EDN (Appendix A). However, analysis according to the development of atopic disease during the first year showed that meconium calprotectin levels were lower in neonates who subsequently developed asthma or CMA compared to those who did not (*p* = 0.02; Wilcoxon test) (Figure 2). Levels of other mediators and cytokines were not associated with the occurrence of an atopic disease (Table 3).

Comparing the different biomarkers according to the presence or absence of an atopic condition, we observed a significant difference at week six for IgE between the presence and absence of cow’s milk allergy and asthma. No significant difference was observed for AD (Appendix A)

#### 3.2.3. Correlation between Biomarkers 

As an expected control, significant correlations were found between weight and height (R = 0.90; *p* < 0.0001), between gestational age and height (R = 0.75; *p* < 0.0001), and between gestational age and weight (R = 0.75; *p* < 0.0001).

Total IgE and tryptase levels were strongly correlated in samples taken at any age. IL-10 and IL-6 were correlated at all ages except at two weeks (Table 4).

#### 3.2.4. Meconium Samples

Tryptase levels were correlated to levels of IL-10 (R = 0.48, *p* = 0.001) and IL-6 (R = 0.46; *p* = 0.001). Tryptase levels were correlated to levels of IgE (R = 0.91, *p* = 0.0001), and strong correlations were observed between levels of calprotectin and IL-1β (R = 0.90; *p* < 0.0001). A negative correlation between total protein concentration and TGF-β (R = −0.36; *p* = 0.01) was observed (Table 3).

#### 3.2.5. Samples at Two Weeks

Strong correlations were observed between levels of calprotectin and IL-1β (R = 0.90; *p* < 0.0001), tryptase and IL-10 (R = 0.88; *p* < 0.0001), and total IgE and IL-10 (R = 0.85; *p* < 0.0001), while total protein concentration and IL-6 were negatively correlated (R = −0.61; *p* = 0.0002) (Table 3).

#### 3.2.6. Samples at Four Weeks

Again, calprotectin and IL-1β were strongly correlated (R = 0.74; *p* < 0.0001). IgE and total protein were also correlated (R = 0.38; *p* < 0.04) (Figure 3).

#### 3.2.7. Samples at Six Weeks

TGF-β was correlated with IL-1β (R = 0.49; *p* < 0.01) and with total proteins (R = 0.43; *p* < 0.03). Total proteins were correlated with IL-1β (R = 0.47; *p* < 0.02) and negatively with IL-6 (R = -0.68; *p* = 0.0003) (Figure 3).

### 3.3. Frequency of Detection of Methanogenic Archaea and Relationship with the Subsequent Development of Atopic Diseases

Using real-time PCR with 16S rRNA archaeal gene PCR primers, we detected methanogenic Archaea DNA in 30/33 (90%) meconium samples, 27/33 (81%) two-week-old samples, 23/29 (79%) four-week-old samples, and 19/24 (73%) six-week-old samples, respectively. We found no significant difference in the frequency of detection nor in CTs according to age at sampling (Table 4).

### 3.4. Unsupervised Analysis of Immunological Markers and Methanogenic Archaea Atthe Neonatal Period, and the Subsequent Occurrence of AD, Asthma and CMA during the First Year

We performed unsupervised analysis of the immunological data, CT of Archaea, and the clinical information of the occurrence of allergic events during the first year of life. Data were analyzed for each of the four sampling times.

For meconium, calprotectin, EDN, and IL-1β levels were negatively and significantly (*p* < 0.001) correlated (r = −0.64) with subsequent development of AD. Calprotectin, EDN and IL-1β had the largest and most significantly (*p* < 0.01) correlated positive correlation coefficients, which were 0.79, 0.53, and 0.51, respectively. No correlation was observed for the Archaea CT with the other parameters (Figure 4b(A)).

At two weeks, IL-1β (r= 0.88) and calprotectin (r = 0.82) had a strong positive correlation with each other, followed by IL-6 (r = 0.62) and EDN (r = 0.59). These biomarkers were significantly (*p* < 0.001) associated. Archaea CTs had a weak (r < 0.5) but positive and significant (*p* < 0.01) association with the biomarkers IL-1β, calprotectin, IL-6, and EDN (Figure 4b(B)).

At four weeks, later occurrence of AD, CMA, and asthma was positively correlated with calprotectin (r = 0.61), IL-1β (r = 0.58), EDN (r = 0.57), and TGF-β (r = 0.57), and negatively correlated with IL-6 (r = −0.71) and IL-10 (r = −0.61). Calprotectin, IL-1β, and EDN were significantly associated with each other (*p* < 0.001). Archaea CTs were positively associated with calprotectin, IL-1β, IL-6, and EDN with a significant correlation (*p* < 0.001), however, they were inversely correlated with allergic events, although the correlation coefficient was low (Figure 4b(C)).

At six weeks, only IL-6 correlated negatively (r = −0.51) with the other biomarkers and allergic events. IL-1β (r = 0.68), IL-10 (r = 0.67), and AD (r = 0.58) showed the strongest positive correlations, with AD significantly (*p* < 0.001) associated with IL-1β and IL-10. Archaea CTs were weakly correlated (r < 0.5) with IL-1β, calprotectin, IL-6, and EDN. However, allergic events were negatively associated with CT, and AD had almost no correlation (Figure 4b(D)).

## 4. Discussion

In this study, we describe a non-invasive screening method for profiling neonatal immunity and its validation in a preterm neonate cohort as a predictive tool for subsequent development of atopic diseases. The method was also applied to meconium samples, which reflect intrauterine processes and contain almost 1000 identified proteins with important functions [37]. The clinical endpoints of this study were evaluated at the age of one year, while the total duration of cohort follow-up will be three years.

Total IgE was detected in over 90% of the samples, at increasing concentrations with age. Transplacental delivery of allergens and preterm sensitization have long been recognized, possibly inducing sensitization and detectable meconial IgE [38,39]. Transplacental transport of maternal IgE able to sensitize fetal mast cells has been recently demonstrated [40], but its role in neonatal immune defenses or subsequent immune disorders is only speculative.

Addressing mast cell tryptase in meconium and later samples, we found that it was detectable only in a minority of meconium samples and during the first month of life, however, it became a common finding at the end of the neonatal period, represented by samples collected at six weeks. Fecal tryptase and IgE levels were strongly associated at each of the studied time points. These results suggest that gastrointestinal mast cells, as opposed to skin mast cells [40], are mostly recruited postnatally, and mature after birth with IgE levels exerting a positive effect. Conversely, maturity of gastrointestinal mast cell populations might be attained during late pregnancy. Tryptase is a serine protease able of autocrine activation of mast cells and induction of proinflammatory effects such as proteolytic cleavage and activation of PAR2 receptors and inactivation of VIP (Vasoactive Intestinal Peptide), associated with smooth muscle relaxation [41]. Through PAR-2 activation, luminal tryptase can contribute to the dysfunction of the gut epithelial barrier [42]. The presence of tryptase in stool samples has been associated with food allergic diseases, dietary exposure and/or mast cell stimulation or increased intestinal mast cell count [43,44]. In addition, fecal tryptase has also been shown to be associated with inflammatory bowel disease and irritable bowel syndrome [45,46].

Focusing on two secreted biomarkers of innate immune cells, neutrophil-derived calprotectin, and eosinophil-derived EDN, we found that fecal samples at all studied ages contained detectable and stable levels of both biomarkers. The levels measured in our preterm cohort were much lower than the reference values [47], but similar to those reported during the first postnatal month in another preterm cohort [48]. These results suggest that the preterm gut contains small numbers of granulocytes, or that such granulocytes are not activated. Indeed, lower neutrophils were reported in preterm infant cord blood [49]. An association between low levels of fecal calprotectin and adverse health conditions, including obesity and sepsis, by age two has been suggested [48].

Proinflammatory and anti-inflammatory cytokines were detected in the meconium and later samples of preterm infants, with the notable exception of TNF-α which was not demonstrated in any sample. Different temporal patterns were demonstrated: IL-6 levels were higher in the meconium than in later samples, while IL-10 was seldom detected and TGF-β and Il-1β displayed a progressive increase between birth and six weeks. Mostly undetectable IL-10 levels were also reported in a pilot study of fecal biomarkers in preterm infants [8]. Although we did not determine the cellular source of fecal cytokines, a shift in immune cells lining the intestine has been demonstrated for macrophages, with resident fetal macrophages being replaced after birth by bone-marrow-derived macrophages [50]. Macrophage cytokine production, most notably of proinflammatory IL-6 and IL-1β, can be persistently altered by metabolic conditions [51]. The increase in TGF-β levels from birth to six weeks might provide a counter-acting mechanism in a proinflammatory environment. Allergic events (asthma or cow’s milk allergy) and atopic dermatitis were also positively correlated with EDN, IL-1β, IL-10, and IL-6 at four and six weeks. The high production of fecal EDN, IL-1β, and IL-10 during the first weeks of life may therefore be an indicator for later risk of allergic diseases.

The neonatal period is paramount for the establishment of the intestinal microbiota. Intrauterine life is associated with low levels of maternal microbial translocation [52]. However, we have recently demonstrated the presence of the viable methanogenic Archaea *Methanobrevibacter smithii* in the meconium, suggesting intrauterine colonization of the fetus by this microorganism [26]. Here, we provided evidence for postnatal persistence of methanogenic Archaea in fecal samples and suggest a possible role in the orientation of intestinal immunity, supported by the negative association between Archaea abundance (inversely proportional to CT values) and the concentrations of EDN, IL-1β, and IL-6. We also found that Archaea abundance at four and six weeks was positively associated with later occurrence of allergic events. Archaea have been shown to produce SCFA which induce regulatory T cell differentiation, downregulate proinflammatory cytokines, and may protect against the occurrence of atopic conditions [19,53,54,55,56,57] However, in a cohort study, the protective effect of methanogenic Archaea was restricted to the species *Methanobrevibacter stadtmanae* [58]. A decrease in the load of beneficial methanogenic Archaea during the first years of life could therefore favor the occurrence of allergic events during the first years of life.

A third line of contribution to protection or increased risk of developing atopic conditions is the genetic background. As an example, a del/del genotype (−2549 −2567 del18) of Vascular Endothelial Growth Factor (VEGF) has been associated with asthma occurrence and irreversible bronchoconstriction [59].

The strengths of our study are methodological and medical:(1)miniaturization and standardization, using small quantities of stool (1 g) and small volumes of extraction buffer (2 mL). The dilution of the samples was corrected by the freeze-drying process, as the lyophilizates were contained in 1 mL of buffer.(2)prevention, thanks to the use of protease inhibitors, of the risk of potential contamination of the handler.(3)suitability for a microarray platform yielding patterns of immune responses rather than individual measurements.(4)suitability for combined immune and microbiological assessment.(5)proof of concept of the immune profiling of fecal mediators in meconium and neonatal samples as predictors of later development of atopic disorders.(6)proof of concept for non-invasive investigation of the immune status of preterm neonates.

The main weakness of this study is the lack of microbiological data outside Archaea. Further studies are warranted for longitudinal immuno-microbiological profiling of meconium and neonatal samples, in preterm and at-term infants. Its validation as a non-invasive diagnostic method will be in line with the currently unmet needs in terms of non-invasive diagnosis of allergy.

## 5. Conclusions

This study allowed us to highlight the presence of mediators in the meconium and feces of preterm infants. We provide proof of concept of the feasibility and value of a standardized fecal mediator assay for non-invasive profiling of neonatal immunity. Such assays can be used for early characterization of the immune status of a newborn. Technical optimization for a multiplex assay could facilitate the implementation of fecal immune profiling in clinical and research laboratories. We report evidence of a correlation between the meconial and neonatal load of methanogenic Archaea and selected fecal biomarkers, and later occurrence of atopic conditions in preterm children.

## Figures and Tables

**Figure 1 jcm-11-02749-f001:**
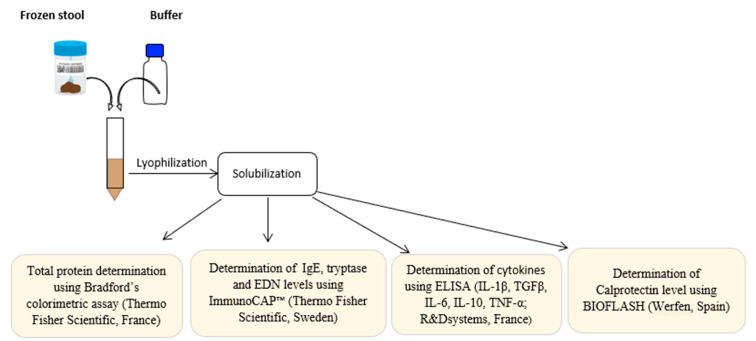
Fecal extraction protocol.

**Figure 2 jcm-11-02749-f002:**
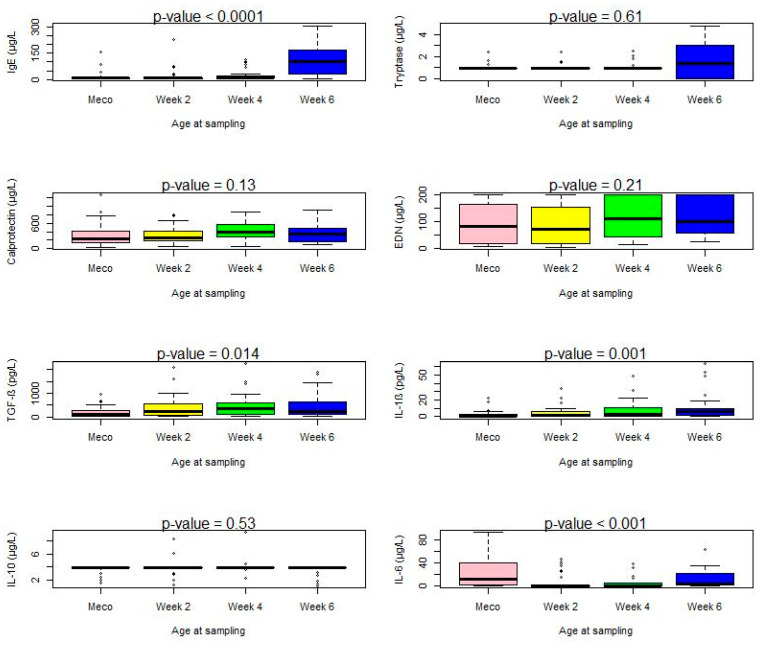
Statistical test: chi-square (frequency), Kruskal–Wallis (concentration). NC, not calculable (calprotectin and EDN were detectable in all samples and at all sampling times).

**Figure 3 jcm-11-02749-f003:**
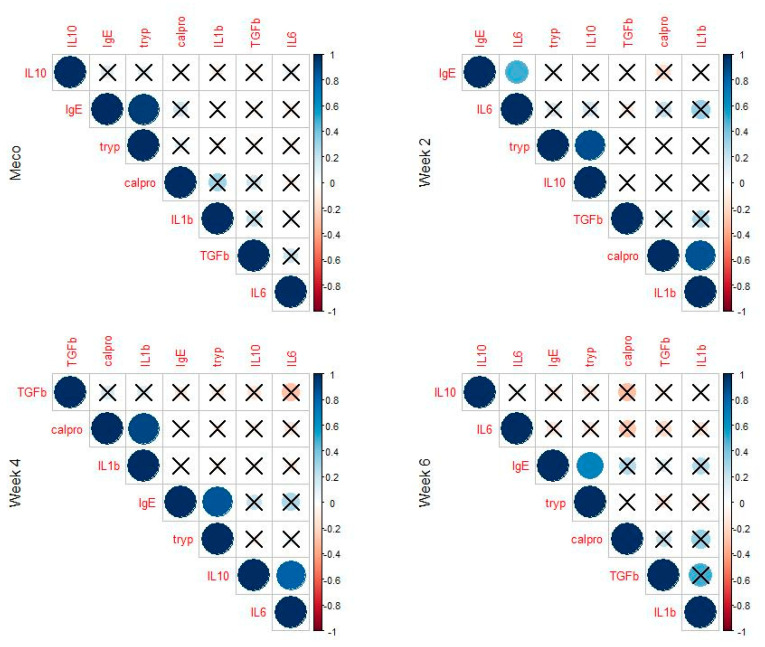
Correlation figure of fecal immune biomarkers.

**Figure 4 jcm-11-02749-f004:**
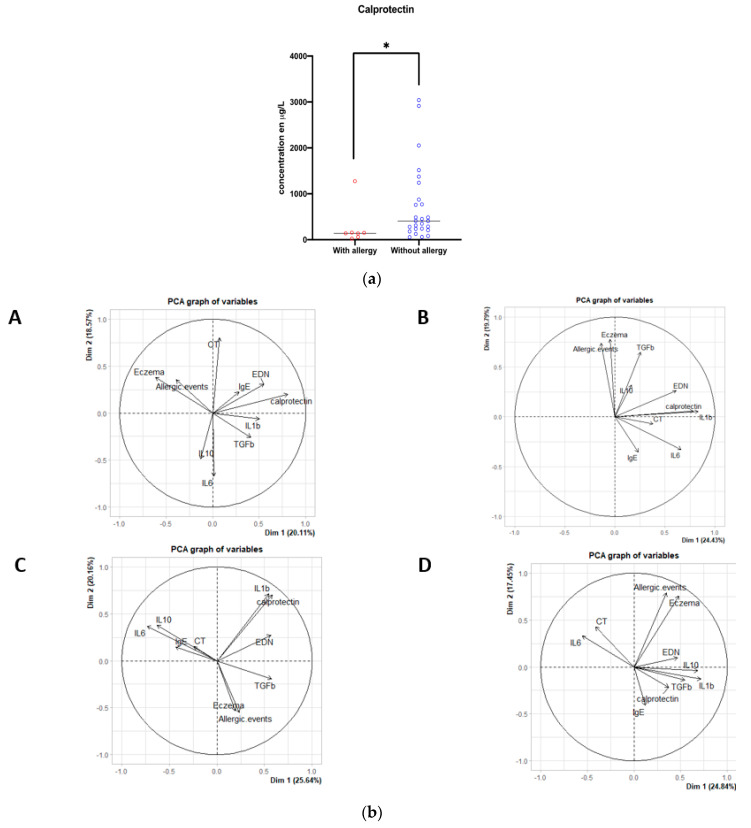
(**a**). Comparison of meconial calprotectin concentration according to the later occurrence of asthma or cow’s milk allergy. (**b**). Principal component analysis of neonatal immune and archaeal biomarkers as a function of later occurrence of atopic conditions. (**A**) Negative and significant (*p* < 0.001) correlation (r = −0.64) between the occurrence of atopic dermatitis with calprotectin, EDN, and IL-1β. Strong positive and significant correlation (*p* < 0.01) between calprotectin (r = 0.79), EDN (0.53), and IL-1b (0.51). No correlation between Archaea TCs and other parameters. (**B**) Correlation between allergic events (asthma or cow’s milk allergy) and atopic dermatitis. Strong positive and significant (*p* < 0.001) correlation between IL-1b (r = 0.88) and calprotectin (r = 0.82), positive correlation between IL-6 (r = 0.62) and EDN (r = 0.59), significant (*p* < 0.001). Low positive (r < 0.5) and significant correlation between Archaea TCs and biomarkers IL-1b, calprotectin, IL-6, and EDN. (**C**) Positive and significant correlation (*p* < 0.001) between atopic dermatitis, allergic events with the markers calprotectin (r = 0.61), IL-1b (r = 0.58), EDN (r = 0.57), and TGF-β (r = 0.57). Negative correlation between IL-6 (r = −0.71) and IL-10 (r = −0.61) with atopic dermatitis and allergic events. Positive and significant correlation between Archaea TCs with calprotectin, IL-1b, IL-6, and EDN. Negative correlation between Archaea Ct and allergic events. (**D**) Negative correlation between IL-6 (r = −0.51) with other biomarkers and allergic events. Strong positive and significant correlation (*p* < 0.001) between IL-1b (r = 0.68), IL-10 (r = 0.67), and atopic dermatitis (r = 0.58). Low correlation (r < 0.5) between Archaea TCs with IL-1β, calprotectin, IL-6, and EDN. Negative correlation between allergic events and TC of Archaea.

**Table 1 jcm-11-02749-t001:** Clinical data for preterm infants investigated for the presence of fecal biomarkers. VD: vaginal delivery; CMA, cow’s milk allergy; CS: cesarean section.

Code	Meconium (M)	Two-Weeks (W2)	Four-Week (W4)	Six-Weeks (W6)	Peripartum Maternal Antibiotic Therapy	Mode of Delivery	Gestational Age	Weight	Size	Asthma or CMA	Atopic Dermatitis
**1**	0	W2	W4	W6	No	VD	30	1275	37	Yes	Yes
**2**	M	W2	W4	W6	Yes	CS	27	925	34	Yes	No
**3**	M	W2	W4	W6	No	CS	26	565	31	Yes	No
**4**	0	0	W4	0	No	VD	25	820	34	Yes	No
**5**	M	W2	W4	0	No	CS	32	1260	39	Yes	No
**6**	M	W2	W4	W6	Yes	CS	27	680	31	Yes	No
**7**	M	0	W4	W6	No	CS	29	1565	42	No	No
**8**	M	W2	W4	W6	No	CS	28	890	33	No	No
**9**	0	0	0	W6	Yes	CS	30	1150	38	No	No
**10**	M	W2	0	0	No	CS	31	1570	43	No	No
**11**	M	W2	W4	0	No	CS	32	1575	44	No	No
**12**	M	W2	W4	0	No	CS	30	1360	39	No	Yes
**13**	M	W2	0	W6	No	CS	25	870	34	Yes	Yes
**14**	M	W2	W4	0	No	CS	32	1155	39	Yes	No
**15**	0	W2	W4	W6	No	CS	25	440	28	No	No
**16**	0	0	0	W6	No	VD	30	1590	41	No	No
**17**	M	W2	W4	W6	No	CS	24	530	31	No	No
**18**	M	W2	0	0	No	CS	26	925	35	No	No
**19**	M	W2	W4	0	No	VD	30	1480	41	No	No
**20**	M	W2	W4	0	No	VD	30	1460	38	No	No
**21**	M	W2	W4	W6	No	CS	29	880	35	No	No
**22**	0	W2	W4	W6	No	CS	28	840	35	No	No
**23**	M	W2	W4	0	Yes	VD	30	1670	43	No	No
**24**	M	W2	W4	W6	No	CS	31	1120	38	No	No
**25**	0	W2	0	0	No	CS	28	915	36	No	No
**26**	M	0	W4	W6	No	CS	26	925	35	No	No
**27**	M	W2	W4	W6	Yes	CS	30	1335	39	No	No
**28**	M	W2	W4	W6	Yes	CS	30	1355	47	No	No
**29**	M	W2	0	W6	No	CS	30	1480	39	No	No
**30**	0	W2	W4	W6	Yes	CS	28	1010	35	No	No
**31**	M	W2	W4	W6	No	CS	29	1050	39	No	No
**32**	M	W2	W4	0	No	CS	29	1190	38	No	No
**33**	0	W2	W4	W6	No	CS	30	1175	39	No	No
**34**	M	0	0	0	No	CS	32	1930	44	No	No
**35**	0	W2	W4	0	Yes	CS	29	1430	30	No	No
**36**	M	W2	W4	0	No	CS	30	1750	43	No	No
**37**	M	0	0	0	No	CS	27	600	29	No	No
**38**	M	W2	W4	W6	No	VD	25	750	32	No	No
**39**	M	W2	0	W6	No	CS	31	980	36	No	No
**40**	M	W2	0	W6	No	CS	31	1410	39	No	No
**41**	M	0	0	0	No	CS	30	770	33	No	No
**42**	M	0	0	0	Yes	VD	32	1568	41	No	No
**43**	M	0	0	0	No	CS	30	1680	39	No	No

**Table 2 jcm-11-02749-t002:** Determination of fecal immune biomarkers.

	Meconium		Two Weeks		Four Weeks		Six Weeks			
	n = 33		n = 33		n = 29		n = 24			
	n (%) Detectable	Median IQR	n (%) Detectable	Median IQR	n (%) Detectable	Median IQR	n (%) Detectable	Median IQR	*p*-Value (Frequency)	*p*-Value (Levels)
Total Proteins (g/L)	33 (100)	9.18 (4.51–13.54)	33 (10,055)	5. 4.23–6.05)	29 (100)	4.53 (3.00–5.52)	24 (100)	6.46 (5.39–7.76)	NS	0.10
Total IgE (µg/L)	30 (90.90)	7.3 (6.4–9.9)	32 (97)	8.47 (6.8–9.8)	27 (93.10)	9.74 (3.39–0.26)	24 (100)	115.08 (41.00–193.70)	0.41	<0.0001
Tryptase (µg/L)	3 (9.1)	<1	3 (9.1)	<1	4 (13.79)	<1	14 (58.33)	1.8 (0.0–3.4)	<0.0001	0.61
Calprotectin (µg/L)	33 (100)	310.4 (151.1–771.3)	33 (100)	291.23 (189.41–487.87)	29 (100)	402.44 (300.06–607.3)	24 (100)	422.37 (335.53–823.30)	NC	0.13
EDN (µg/L)	33 (100)	83.2 (19.3–165.0)	33 (100)	70.1 (17.8–152.5)	29 (100)	109.0 (44.2–200.0)	24 (100)	98.1 (57.5–200.0)	NC	0.21
TGF-β (pg/L)	24 (72.7)	121.3 (4.6–258.9)	30 (91)	267.43 (61.71–1000)	26 (89.65)	384.57 (129.60–936)	22 (91.66)	466 (104.36–1430.29)	0.09	0.014
IL-1β (pg/L)	13 (39.4)	0.12 (0.1–2.7)	28 (84.8)	1.53 (0.37–6.53)	25 (86.20)	3.27 (0.31–10.76)	22 (91.66)	6.23 (1.66–20.84)	<0.0001	0.001
IL-10 (pg/L)	4 (12.12)	3.9 (3.9–3.9)	6 (18.18)	3.9 (3.9–3.9)	5 (17.24)	3.9 (3.9–3.9)	5 (20.83)	3.9 (3.9–3.9)	0.85	0.53
IL-6 (pg/L)	25 (75.75)	11.6 (0.5–43.7)	7 (21.21)	0.2 (0.2–0.2)	20 (68.96)	0.2 (0.2–0.2)	20 (83.33)	3.77 (1.66–19.25)	<0.0001	<0.001

Concentrations are expressed as median and interquartile ranges (IQR). n (%): number of samples in which the biomarker was detected (relative frequency of detection). The median and IQR were calculated by restricting the results above the lower LOQ (limit of quantitation) for each analyte. Statistical test: chi-square (frequency), Kruskal–Wallis (concentration). NC, not calculable (calprotectin and EDN were detectable in all samples and at all sampling times).

**Table 3 jcm-11-02749-t003:** Comparison of mediators and cytokines and the occurrence or absence of an atopic condition between years 0 and 1. Statistical test used: Wilcoxon test.

Variables	Allergic Condition (APLV and Asthma)	*p*-Value			
		Meconium	2 Weeks	4 Weeks	6 Weeks
IgE	Yes	0.27	0.06	0.12	0.03
	No				
Calprotectin	Yes	0.27	0.18	0.91	0.61
	No				
EDN	Yes	0.59	0.19	0.41	0.87
	No				
TGF-β	Yes	0.09	0.18	0.76	0.76
	No				
IL-1β	Yes	0.37	1.00	0.28	0.91
	No				
IL-10	Yes	0.62	1.00	0.89	0.13
	No				
IL-6	Yes	0.61	0.28	0.37	0.75

**Table 4 jcm-11-02749-t004:** Result of the detection of methanogenic Archaea.

	Meconium (n = 33)		Two Weeks (n = 33)		Four Weeks (n = 29)		Six Weeks (n = 26)		*p*-Value (Frequency)	*p*-Value (CT)
	n (%)	Median IQR	n (%)	Median IQR	n (%)	Median IQR	n (%)	Median IQR		
CT qPCR	30 (90.9)	36.74 (33.85–38.24)	27 (81.81)	37.20 (36.07–38.33)	23 (79.31)	37.75 (36.13–38.50)	19 (73.03)	38.28 (37.27–39.96)	0.34	0.12

CT methanogenic Archaea are expressed as median and interquartile ranges (IQR). n (%): number of samples in which methanogenic Archaea were detected (relative frequency of detection). The stool concentration factor and median and RDI were not included in our calculations, and the median and RDI were calculated by restricting the results above the lower LOQ (limit of quantitation) for each analyte. Statistical test: Kruskal–Wallis.

## Data Availability

Not applicable.

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
