# Peer review of "A Non-Invasive Neonatal Signature Predicts Later Development of Atopic Diseases"

_jcm, 2022, doi:10.3390/jcm11102749_

Round 1

Reviewer 1 Report

Congratulations on the idea of work and extensive presentation of the results!

The article presented for review raises an important aspect of quick and accessible allergy diagnostics. It seems quite complicated for everyday diagnostics schedule, especially outside the hospital's department of pediatrics or allergology, but the conclusions reached by the authors are valuable.

From the reviewer's point of view, I draw attention to the need to correct spelling errors in the text.

I think that in the Discussion it is worth adding information whether observations in atopic children will be continued in the following years of life and on larger group of patients, or whether the study has already been definitively completed.

It can be noted whether there are reports in the available literature about similar stool studies, both in children and adults.

The authors focused on the determination of markers in the aspect of atopic dermatitis and cow's milk allergy - the work will gain in value if the revised version also mentions other studies of markers of allergic diseases, e.g. bronchial asthma (one can quote the article "The -2549 -2567 del18 polymorphism in VEGF and irreversible bronchoconstriction in asthmatics" J.Investig.Allergol.Clin.Immunol. 2019 Vol.29 no.6 pp.431-435).

Author Response

Dear professor, thank you for your suggestion,
Thank you for your suggestion. We have revised the paper taking into account all our suggestions. We have added the reference you suggested, and we have also mentioned that the cohort study continues until 3 years of life. 

Reviewer 2 Report

Although the theme itself is interesting (the association of neonatal gut microbiome and the risk of atopic disease), the article is hard to read as it is rather redundant and lengthy. 

In the table 3, the variables were not significantly different between the groups with or without allergic events. In the table 4, the immune biomarkers were correlated to each other (whether the presence of atopic disease is correlated to these marker is not clarified in this part). Also, the CTs of methanogenic Archaea did not differ. However, all of sudden, in an unsupervised analysis of immunological data, calprotectin, EDN and IL-1β levels were negatively and significantly (p < 0.001) correlated (r = -0.64) with subsequent development of AD. And later, the authors repeated the correlation between the biomarkers, which occupied the most part of significant findings. However, I think the point of the present study is rather "correlation between biomarkers and atopic risk" than "correlation between immune biomarkers". 

I cannot understand and follow the logical process of the conclusion.

Additionally, the authors mostly used tables to present their data, but you may consider using graphs or variable ways to efficiently and clearly deliver the meaning of your data.  

Author Response

Dear professors, thank you for your suggestion,
Thank you for your suggestion. We have revised the paper taking into account all our suggestions. Table 3 is just a comparison of the data according to the presence or absence of an allergic pathology. The main component analysis is correlation of biomarkers and TCs of archaea according to the presence of an allergic disease. 
You will find in this new version the figures you asked us for. 

Round 2

Reviewer 2 Report

Thank you for your work.